



# Estimates of climatic influence on the carbon cycle

Ian Enting[1] and Nathan Clisby[2]

[1]CSIRO Climate Science Centre, Oceans and Atmosphere, Aspendale, Vic, Australia
[2]Department of Mathematics, Swinburne University of Technology, PO Box 218, Hawthorn Vic, 3122, Australia

**Correspondence:** Ian Enting (ian.g.enting@gmail.com)

**Abstract.** The influence of climatic change on the carbon cycle is important as part of a $CO_2$-climate feedback loop. However the magnitude of the coupling depends on the timescales involved. We expand on previous analyses of the ice-core $CO_2$ data from the pre-industrial period 1000–1750, extending the analysis into the 20th century. Our results emphasise the limitations of characterising the climate-to-$CO_2$ influence by a single number $\gamma$. Even once a time-scale dependence is incorporated, the

coldest part of the Little Ice Age seems to reflect different behaviour to that in earlier or later centuries. Different temperature reconstructions appear to capture distinct aspects of pre-industrial climate fluctuations that lacked global coherence. An exploratory study extends the analysis into the industrial period. In this study, most paleo-temperature data fail to fit the plateau (or plateaus) in 20th century ice-core $CO_2$, with one particular reconstruction as an exception. One interpretation of this fit is that although the reconstruction does not closely reflect hemispheric temperature changes, it samples a pattern of variation

where the terrestrial carbon exchange is anomalously sensitive to regional climate variations. These various results suggest that this type of empirical study may have limited applicability to the 21st century.

## 1 Introduction

The extent to which changing climate induces changes in the carbon cycle is important for two primary reasons. Changes in the

carbon cycle are important as direct impacts of climate change, and feedbacks in the coupled climate-carbon system have the potential to amplify changes in climate. See, for example, Denman et al. (2007) and chapter 29 of Bonan (2008) for reviews of the coupled climate-carbon system.

We follow Rubino et al. (2016) in using a time-scale dependent generalisation of the (un-numbered) equation from Friedlingstein et al. (2003) which gives $Q(t)$, the change in atmospheric carbon mass, in response to anthropogenic emissions $S(t)$, and

temperature changes, $\Delta T$, as:

$$Q(t) = \frac{\int S(t)\,dt}{1+\beta_O+\beta_L} - \frac{(\gamma_O+\gamma_L)\Delta T}{1+\beta_O+\beta_L} \tag{1}$$



where $\beta_O$ and $\beta_L$ specify how the carbon cycle processes partition fluxes between atmosphere, oceans and land biosphere in the ratio $1 : \beta_O : \beta_L$ (Oeschger et al., 1980). $\gamma_O$ and $\gamma_L$ specify how temperature changes contribute to emissions. Our present analysis is in terms of aggregated quantities $\beta = \beta_O + \beta_L$ and $\gamma = \gamma_O + \gamma_L$.

Relation (1) has provided a framework for characterising climate-to-carbon influence both from empirical estimates and modelling studies (e.g. Friedlingstein et al., 2006; Gregory et al., 2009).

The outline of the paper is as follows:

Section 2 gives the defining equations for a linearised representation of carbon-climate interactions and shows, using a Laplace transform analysis, that this defines a time-scale dependent generalisation of relation (1). Section 3 addresses the specifics

of estimating the effect of temperature on the carbon cycle. The two important aspects are: (i) what is being estimated? and (ii) how is the estimation done? Section 4 presents our estimates, extending and refining the pre-industrial cases considered by Rubino et al. (2016) and extending the analysis to the 19th and 20th centuries. Section 5 considers the significance of our results, in particular for the fit based on the paleo-temperatures from Moberg et al. (2005). This reconstruction has poor agreement with direct observations of northern hemisphere temperatures since the mid 19th century, but leads to a fit that

reproduces hard-to-explain plateaus in the 20th century $CO_2$ record. A possible interpretation is that rather than reflecting hemisphere-wide temperature changes, the Moberg et al. reconstruction represents a pattern of variation where the terrestrial carbon exchange is anomalously sensitive regional to climate variations. The concluding remarks describe our results in the context of other interpretations attributing past $CO_2$ changes to human activity (e.g. Kaplan, 2015; Bastos et al., 2016) and discusses the implications for changes over the 21st century. Details of the notation are given in an appendix. A further appendix

discusses disaggregation of the global-scale relations as a basis for extending the analysis to allow for regionally diverse climatic fluctuations.

## 2    Linear response analysis

The most general form of causal linear relations can be expressed in terms of responses given by convolutions between forcing influences with impulse response functions. Such integro-differential relations can be conveniently expressed using Laplace

transforms since convolutions in the time domain become products of transforms in the frequency domain.

Enting (2010) analysed the carbon-climate feedback loop in terms of linear response relations for climate, expressed as temperature anomaly, $W(t)$ (with transform $w(p)$)

$$w(p) = u(p)[\alpha q(p) + f_{\text{other}}(p)] \qquad (2)$$

where $U(t)$ (with transform $u(p)$) gives the temperature response to radiative forcing. The radiative forcing is partitioned

into that from $CO_2$ (linearised as $\alpha Q(t)$, with transform $\alpha q(p)$), and other radiative forcing $F_{\text{other}}(t)$ with transform $f_{\text{other}}(p)$. However, for the purposes of the present analysis, it is preferable to consider temperature changes as

$$w(p) = \alpha\, u(p)\, q(p) + w_{\text{other}}(p) \qquad (3)$$





For atmospheric carbon, the response relation uses an impulse response function $R(t)$ (with transform $r(p)$) to express $Q(t)$ (with transform $q(p)$), the excess of $CO_2$ over pre-industrial levels as:

$$q(p) = r(p)\left[s(p) + h(p)w(p)\right] \tag{4}$$

where the response function $R(t)$ gives the proportion of emissions remaining in the atmosphere after time $t$. The emissions are the sum of anthropogenic emissions, $S(t)$ (with transform $s(p)$), and emissions driven by temperature changes $W(t)$. These latter emissions are represented as a linear response described by response function $H(t)$ (with transform $h(p)$).

Defining $\gamma(p) = -h(p)/p$, and noting that $s(p)/p$ is the integrated emissions and $pr(p)$ is the airborne fraction $\psi(p) = 1/(1+\beta(p))$ (Enting, 1990), relation (4) becomes

$$q(p) = \frac{s(p)/p}{1+\beta(p)} - \frac{\gamma(p)w(p)}{1+\beta(p)} \tag{5}$$

This represents a time-scale dependent generalisation of (1).

From (2) and (4), the feedback loop has a gain

$$g(p) = \alpha u(p)r(p)h(p) = -\alpha u(p)\psi(p)\gamma(p) \tag{6}$$

with forcing of either carbon or climate being amplified (Gregory et al., 2009; Enting, 2010) by a factor

$$\frac{1}{1-g(p)} = \frac{1}{1-\alpha u(p)r(p)h(p)} = \frac{1}{1-\frac{\alpha u(p)\gamma(p)}{1+\beta(p)}} \tag{7}$$

Thus we define a $CO_2$ response that includes feedbacks as

$$r_{\text{FB}}(p) = \frac{r(p)}{1-\alpha u(p)r(p)h(p)} = \frac{1/p}{1+\beta(p)-\alpha u(p)\gamma(p)} \tag{8}$$

If the $\beta$ contribution is regarded as a negative feedback (or sum of feedbacks) on atmospheric $CO_2$ accumulation (i.e. the $1/p$ factor corresponds to integration), equation (8) illustrates the principle that linear feedbacks add (e.g. Gregory et al., 2009).

Inserting (3) into (4), and using (8), gives

$$q(p) = r_{\text{FB}}(p)\,s(p) + r_{\text{FB}}(p)\,h(p)\,w_{\text{other}}(p) \tag{9}$$

In the absence of any ability to partition $w(p)$, relation (9) can only be used for empirical studies in the pre-industrial period with $s(p) = 0$, but the relation may be useful for characterisation of modelling studies. Our representation (4) is essentially equivalent to that used by Bauska (2013), except that he worked in terms of fluxes rather than concentrations. Flux relations are obtained by dividing relation (4) by $r(p)$, since $1/r(p)$ represents a deconvolution operation (Enting and Mansbridge, 1987).

While the Laplace transform greatly simplifies formal manipulation of integro-differential relations, for practical calculations we need to express the relations in the time domain.



We express (4) as

$$
\quad Q(t) = \int_0^t dt' \, R(t-t') \, S(t')
$$

$$
+ \int_0^t dt' \, R(t-t') \int_0^{t'} dt'' \, H(t'-t'') \, W(t'') \tag{10}
$$

with the time origin taken sufficiently far in the past that the influence of initial conditions can be neglected except for their influence on the reference levels for anomalies $Q(t)$ and $W(t)$.

For $CO_2$, the response function $R(t)$ has been studied extensively because of its role in defining Global Warming Potentials, and the use of these GWPs in emission reporting and target setting. Joos et al. (2013) report the results of a recent intercomparison. The response functions presented by Joos et al. (2013) explicitly include the temperature feedback, and so in the present notation, they correspond to $R_{FB}$. However, even estimated response functions that do not include temperature feedbacks explicitly, can include them implicitly if they are calibrated against 20th century $CO_2$. This was noted by Enting (2011), and illustrated with an example where a model with no explicit feedback gave projected concentrations within the range of values from an upgraded version of the model that did include temperature feedbacks.

In terms of equation (4), $p\,r(p)$ specifies the airborne fraction as a proportion of total sources, including those from feedbacks, However, when considering only anthropogenic sources, the $1/(1-g)$ amplification will lead to an effective airborne fraction given by $\psi_{eff} = p\,r_{FB}(p)$. Recognising the ambiguity between $r(p)$ and $r_{FB}(p)$ plays a significant role in our analysis. Equally important is the recognition that estimates of $R(t)$ for use in contemporary GWP calculations (e.g. Joos et al., 2013) generally calculate responses for perturbations about $CO_2$ concentrations from around the end of the twentieth century at which time various non-linearities led to higher values of the response.

In contrast, much less is known about the functional form of $H(t)$, the aggregate response of $CO_2$ fluxes to temperature changes. For empirical data-based studies, simple functional forms provide a pathway to exploratory analysis.

We explore the parameterisation:

$$
\quad h(p) = \gamma' \frac{p}{p+1/\tau} = \gamma' \left[ 1 - \frac{1/\tau}{p+1/\tau} \right] \tag{11}
$$

or equivalently

$$
\gamma(p) = -\gamma' \frac{1}{p+1/\tau} \tag{12}
$$

Here, $\tau$ is a representative relaxation time and $\gamma'$ is a factor that specifies the overall scale of the climate-to-carbon influence.

This provides a simple functional form that can be used to explore empirical climate-to-carbon influence. Approximation (11) interpolates between the two extremes of constant $h$ and constant $\gamma$:

$$
h(p) \approx \gamma' \quad \text{and} \quad \gamma(p) \approx -\gamma'/p \qquad \text{for large } p, \tag{13}
$$

while

$$
h(p) \approx p\gamma'\tau \quad \text{and} \quad \gamma(p) \approx -\gamma'\tau \qquad \text{for small } p \tag{14}
$$




Inverting the Laplace transform (11) gives

$$H(t) = \gamma' \left[ \delta(t) - \exp(-t/\tau)/\tau \right] \tag{15}$$

The second term in (15) is a re-adjustment of the terrestrial system. It is not a response to atmospheric $CO_2$ change and so in terms of the framework used by Friedlingstein et al. (2003) (equation 1 above) it is part of $\gamma$ rather than $\beta$. The time integrated flux $\int^t H(t')\, dt'$ from an instantaneous temperature perturbation is a step function increase (of size $\gamma'$ at time zero) followed by exponential decay to zero.

For the purposes of estimation, we define

$$H(t) = \gamma' H_{\text{ref}}(t) \tag{16}$$

and transform temperature data using $H_{\text{ref}}(t)$, leaving $\gamma'$ as a factor to be estimated from observational data.

## 3 Estimation

As with Rubino et al. (2016), our analysis is based on regression fits of equation (10) with scale factors, $\theta_\gamma$ and $\theta_s$, introduced to enable us to estimate the scale factor $\gamma'$ and the normalisation of $R_{\text{ref}}$.

$$
\begin{aligned}
Q(t) = {} & \theta_0 + \theta_s \int_0^t dt'\, R_{\text{ref}}(t-t')\, S(t') \\
& + \theta_\gamma \int_0^t dt'\, R_{\text{ref}}(t-t') \int_0^{t'} dt''\, H_{\text{ref}}(t'-t'')\, W(t'')
\end{aligned}
\tag{17}
$$

where the offset $\theta_0$ reflects the fact that the notional time of pre-industrial equilibrium ($Q(t) = 0$) may not match the time of the zero temperature anomaly. We adopt the common statistical notation of using the 'hat' symbol to indicate estimates of the various $\theta$.

Within this general framework, there are a number of choices, with specific cases listed in Section 4. These include: the choice of $\tau$, the choice of paleo-temperature data sets and the choice of the time period fitted.

Finally, the choice of response function $R_{\text{ref}}$ has to address whether it is regarded as an estimate of $R(t)$ or an estimate of $R_{\text{FB}}(t)$. We have explored the use of several different response functions: $R_{\text{init}}$ from Enting et al. (1994); $R_{\text{Bern}}$ model (as used for GWP definition in IPCC AR4 report (p213)); $R_{\text{Joos}}$ Table 5 from Joos et al. (2013); $R_{\text{FB}}$ from Joos et al. (2013) (table S2 of supplementary information); $R_{\text{no-FB}}$ from Joos et al. (2013) (table S2 of supplementary information). Figure 2 compares values of $pr(p)$ (i.e. the airborne fraction) for various response function estimates. These loosely fell into three categories: with response functions labelled Init and no-FB forming one pair, Bern and the explicit FB another pair, while the Joos13 response function is somewhat distinct. All the results presented here use $R_{\text{init}}$.

These considerations give rise to various forms of the regression, described in equations (18), (19) and (20), expressed in terms of the Laplace transform for compactness of notation.



If model-based estimates of $r(p)$ are used for $r_{\text{ref}}$ then the appropriate regression for the pre-industrial case is

$$q(p) = \theta_0/p + \theta_\gamma r_{\text{ref}}(p) h_{\text{ref}}(p) w(p) \tag{18}$$

These give $\hat{\theta}_\gamma$ as an estimate of $\gamma'$, conditional on assuming that $r_{\text{ref}}$ is a correct estimate of $r(p)$. If $r_{\text{ref}}(p)$ corresponds to $r_{\text{FB}}(p)$ rather than $r(p)$, then $\hat{\theta}_\gamma$ is actually an estimate of $\gamma' \times (1-g)$. This potential correction was neglected by Rubino et al. (2016) who used a response function estimate from Joos et al. (2013).

For the industrial period, relation (4) would imply

$$q(p) - r_{\text{ref}}(p) s(p) = \theta_0/p + \theta_\gamma r_{\text{ref}}(p) h_{\text{ref}}(p) w(p) \tag{19}$$

with $r_{\text{ref}} = r(p)$. However given the ambiguities regarding the response function, we have not analysed this case.

If $r_{\text{ref}}(p)$ corresponds to $r_{\text{FB}}(p)$, then the industrial period needs to be analysed in terms of the time-domain equivalent of

$$q(p) = \theta_0/p + \theta_s r_{\text{ref}}(p) s(p) + \theta_\gamma r_{\text{ref}}(p) h_{\text{ref}}(p) w(p) \tag{20}$$

Thus $1/\hat{\theta}_s \approx r_{\text{FB}}(p)/r(p)$ is (subject to caveats above) an estimate of the loop amplification, $1/(1-g)$. It also follows that, if $\gamma(p)$ is to be defined in terms of $r(p)$ (i.e. if we parallel the definition from Friedlingstein et al. (2003)), then the scale factor should be $\hat{\theta}_\gamma/\hat{\theta}_s$.

However, if the non-$CO_2$ warming, $W_{\text{other}}$, is small (or of a similar time-dependence to $Q(t)$) then a comparison of (4) and (9) suggests that the data might be fitted without explicit reference to the temperature, giving $\theta_\gamma \approx 0$ (and $\theta_s \approx 1$ if the response function has been calibrated to fit 20th century $CO_2$). This will also occur if the temperature reconstruction cannot be matched to the $CO_2$ changes.

## 4 Estimates

The data sets used as input to these regression analyses are the same as were used in the initial application by Rubino et al. (2016). The $CO_2$ data were concentrations measured in bubbles from the DSS and DE08 ice cores from Law Dome, Antarctica (MacFarling Meure et al., 2006; Rubino et al., 2013). The data consist of 261 concentrations for years from 154 to 1974. The temperature data are four paleo-reconstructions given as annual values:

**M2005**: The data from Moberg et al. (2005) is for the northern hemisphere and spans the years 1–1979;

**CL2012**: the data from Christiansen and Ljungqvist (2012) is for north of 30N and spans the years 1–1973;

**PAGES Arctic**: The PAGES Arctic data (PAGES 2k network, 2013) span the years 1–2000; and

**Mann99**: the Mann et al. (1999) northern hemisphere data span the years 500–2000.

Since this last record starts more recently than the others, the transformed data can only be fitted from later start dates. These reconstructions are plotted in Figure 1, shown as 20-year running means, to reduce visual clutter. They show a number of differences which translate into differences in our estimates, as discussed below.

We have extended the cases reported by Rubino et al. (2016) using equation (18) to fit periods through to 1800. All these cases use $\tau = 100$ years, but unlike Rubino et al. (2016) we use the response function $R_{\text{init}}$ from Enting et al. (1994). Some





results are given in Table 1 and some fits are plotted in Figure 3. In Table 1 cases where the start is within 500 years of the beginning of the temperature data as bracketed to indicate the possible influence of end effects.

Clearly, there are major differences between the estimates, both for different time periods and different temperature reconstructions. One notable trend is that the fits for shorter periods 'explain' more of the change than fits over longer periods. In other words,we can find a range of medium term periods of agreement, but these do not fit together into a consistent long-term pattern. When consistent patterns do occur, a smaller amplitude of change will in general imply a higher estimate of $\hat{\theta}_\gamma$. The diverse results suggest that different reconstructions are sampling different aspects of climatic fluctuations which, as emphasised

by Neukom et al. (2019), lack global coherence.

    The estimates $\hat{\theta}_\gamma$ are generally larger than the estimates of $\gamma'$ quoted by Rubino et al. (2016). The direct reason for this is our use of $R_{\mathrm{init}}$ rather than $R_{\mathrm{Joos}}$. The latter response is calculated from perturbations to late 20th century $CO_2$ leading to larger values than the response $R_{\mathrm{init}}$ which is calculated from perturbations to pre-industrial conditions. In addition, as discussed below, our estimates $\hat{\theta}_\gamma$, may well be underestimates of $\gamma'$ to the extent that $R_{\mathrm{init}}$ implicitly includes the effects of

climate-to-carbon feedbacks (Enting, 2011).

    We have extended these analyses by going on to consider the industrial period, using equation (20). We fitted data over the period 1800-1978, as well as composite fits that extend back into the pre-industrial period. Various estimates of $\theta_\gamma$ and $\theta_s$ (denoted $\hat{\theta}_\gamma$ and $\hat{\theta}_s$) are given in Table 2 and the fits given by the composite 1500–1978 are shown in Figure 4.

    From these results we observe:

– since the majority of the fits have $\theta_s$ significantly less than 1, we conclude that, for the reasons discussed by Enting (2011), the response function $R_{\mathrm{init}}$ implicitly includes the effects of feedbacks;

    – the exceptions to $\hat{\theta}_s \approx 1$ are the longer term fits with the C&L and the PAGES Arctic data – we interpret this as an inconsistency between the changes implied by these reconstructions and the observed changes. Consequently all change is being attributed to anthropogenic emissions. We note that these cases are those that produce smaller estimates, $\hat{\theta}_\gamma$,

even when only pre-industrial data are used;

    – more generally the ambiguity between forms (4) and (9) leads to the estimates $\hat{\theta}_s$ and $\hat{\theta}_\gamma$ having large negative correlations;

    – the PAGES Arctic data sets produce comparable estimates, $\hat{\theta}_\gamma$, when fitted over 1800–1978 and 1500–1800 but do not give a consistent fit over 1500–1978, presumably because the short-term fits do not have consistent reference levels;

– the Mann data give broadly similar estimates for fits including the industrial period and pre-industrial fits over 1300–1750 and 1500–1800. Fits involving earlier periods may be reflecting 'end effects' since this record only begins in 500;

    – the Moberg record presents a particularly puzzling case. The estimates $\hat{\theta}_\gamma$ for the pre-industrial period are systematically lower than for the industrial period. These latter estimates give the surprising result of fitting the plateaus in the $CO_2$ record that occur in the first half of the 20th century. Figure 5 shows 20th century detail of the fits.





## 5 Implications

As in the study by Rubino et al. (2016) we find no single fit can describe the pre-industrial $CO_2$ variation. A similar result was found by Frank et al. (2010). Our most striking result is that fitting data over the period 1500–1980 using temperatures from Moberg et al. (2005) could reproduce the plateaus observed in ice core data from the first half of the 20th century. The plateaus appear more strongly when using the larger $\hat{\theta}_\gamma$ obtained when the data are fitted over the period 1800–1978. The reason that this differs from other cases, is that, over the 20th century, the Moberg et al. (2005) reconstruction differs significantly from the others, in ways that do not agree with direct observations of northern hemisphere averages. Figure 7 shows the Mann and Moberg reconstructions along with the northern hemisphere land temperatures from direct observations (Jones et al., 2012). This suggest that the data are sampling a particularly temperature-sensitive set of locations. This would explain why studies such as Rafelski et al. (2009) and Bastos et al. (2016) fail to explain these plateaus. In addition the fit with the Moberg et al. (2005) data also fits the lower $CO_2$ concentrations circa 1600.

The Moberg et al. (2005) reconstruction is a synthesis of high resolution data from tree rings and disparate lower resolution data. The 7 tree ring records comprise 3 from the Pacific north west of America, one from northern Scandinavia and three high latitude Asian sites. The low resolution data are various lake and marine sediments and speleothems, spanning a wider range of latitudes.

Our analysis is based on the proposition that $\gamma$, as used by Friedlingstein et al. (2003), needs to be considered in terms of time-scale dependence. (The analysis of 20th century data, suggests that in some cases, one need also consider spatial dependence as outlined in the Appendix B). Figure 6 plots the time-scale dependence of some of our estimates and compares them to other studies.

## 6 Comparison with other studies

Woodwell et al. (1998) give a range of estimates which should be regarded as $\gamma$ times the appropriate airborne fraction, i.e. estimates of $pr(p)\gamma$ because they do not include the $1/(1 + \beta_O + \beta_L)$ factor in their definition of $\gamma$. The smaller values for contemporary timescales compared to changes over the Little Ice Age are broadly consistent with the curves shown in Figure 6. Keeling et al. (1989) discuss the various possible processes. They estimate 2 ppm/K ($\approx$ 4 PgC/K) based on decadal-scale modulation of ENSO variability.

Cox and Jones (2008) studied $CO_2$ and climate over the LIA using a lagged response to temperature. They fitted LIA changes with 40±20 ppm/K over 1500–1750 using a 50-year lag. They suggests that using lags is comparable to the response to step change. However, our approach where we incorporate such a response explicitly seems to give rather different results.

Frank et al. (2010) generated an ensemble of cases with a range of different temperature reconstructions, calibration periods, lags and data smoothing fitted to ice core $CO_2$. While they denoted their sensitivity factor $\gamma$, it was defined with respect to the atmosphere (i.e. not the Friedlingstein et al. (2003) definition, from which it differs in sign and by a factor corresponding to airborne fraction).





Their estimates (and 1 s.d. range) were $4.3 \pm 3.5$ ppm/K (9.13 PgC/K) for 1050–1549 and 16.1 $\pm$12.5 ppm/K (34.2 PgC/K) for 1550–1800. Their modal value was 8.5 ppm/K (20.3 PgC/K). These estimates convert to $\gamma$ of about $-23$ PgC/K, $-85$ PgC/K and $-51$ PgC/K respectively, assuming an airborne fraction of 0.4.

Bauska et al. (2015) deconvolved $CO_2$ data to find cumulative changes in carbon stocks and then performed regressions against temperature data which had been smoothed with functions $X_k = \lambda_k \exp(-\lambda_k t)$ with decay times of $\lambda_k^{-1} = 25, 50, 75,$ 100, 125 and 150 years.

These fits correspond to

$$-r(p)^{-1} q(p)/p \sim \beta_k \, x_k(p) \, w(p) \tag{21}$$

with $x_k(p) = \lambda_k/(p + \lambda_k)$. This corresponds to fitting $\gamma(p) \, w(p) \sim \beta_k \, x_k(p) \, w(p)$

Thus for the appropriate timescale $p$

$$\hat{\gamma}(p) = \hat{\beta}_k \frac{\lambda_k}{p + \lambda_k} \tag{22}$$

The various estimates $\hat{\beta}_k$ are broadly consistent with them fitting the $1/p = 125$ years case, with $\gamma(p = 0.008) \approx -47$ PgC/K. This is shown as a circled point in Figure 6

Bauska et al. (2015) also attempted multivariate fits of paleo-temperature data from 6 regions, often failing to find statistical significance. The analysis in Appendix B below may provide a way of refining such multi-variate studies. The discussion by Neukom et al. (2019) of the lack of global coherence in pre-industrial climate fluctuations emphasises the need for such a generalisation of our formalism.

## 7    Concluding remarks

We have extended the analysis by Rubino et al. (2016) further back into the pre-industrial period and also into the 19th and 20th centuries. The studies for the industrial period clarify the nature of the estimation for the pre-industrial period. Many of our results indicate that, as suggested Enting (2011), the response functions already include the effect of carbon-climate feedback. Thus our estimates $\hat{\theta}_\gamma$ should be regarded as estimates of $\gamma' \times (1 - g)$. Notionally, $\hat{\theta}_s$ can be considered as an estimate of $1 - g$, but:

— the factor should be regarded as have a timescale dependence, i.e. written as $1 - g(p)$. Comparison of the FB vs no-FB cases shown in Figure 2 indicate a timescale dependence; and

— the correction factor, as a representation of $r(b)/r_{\mathrm{FB}}(p)$ should be the same for all cases. While the different temperature reconstructions may represent different patterns of climate forcing and lead to different estimates of $\gamma'$, the factor $1 - g(p) = r(b)/r_{\mathrm{FB}}(p)$ should be the same for all cases.

We have revisited a number of earlier analyses. We conclude some of the spread in older estimates of climate-to-carbon influence is due to over-parameterisation — failure to consider time-scale dependence. In some cases, notably (Bauska et al., 2015), we have been able to show how their estimates can be related to our formalism.





For particular cases, we find that our estimates of $\gamma'$ can be used to extrapolate the climate to carbon influence outside the time interval for which they were fitted. In spite of this success, we are unable to fit all LIA variation. Kaplan (2015) has

suggested that changes in land use may have been an important influence. In contrast, Rubino et al. (2016) use COS data to support their conclusion that $CO_2$ changes reflect changes in terrestrial carbon storage in response to climate change. Our results suggest that some of the differences in interpretation may be due to regional differences in climatic variation.

The fit to plateaus in 20th century $CO_2$ when using data from Moberg et al. (2005) is obtained because this reconstruction is not representative of northern hemisphere temperatures. It reflects, at least in part, a pattern of temperature variation different

from greenhouse warming. The lack of global coherence identified by Neukom et al. (2019) for preindustrial times remains relevant for the first half of the twentieth century.

In view of our conclusion that the estimation of $\gamma$ reflects regionally disparate climate-to-carbon effects operating over past centuries, we need to clarify the assumptions involved in extrapolating such empirical results to the 21st century:

– it is assumed that northern and southern hemisphere temperatures will track each other more closely in future than over

the early 20th century and so $dM/dT_{\mathrm{NH}}$ and $dM/dT_{\mathrm{global}}$ are essentially equivalent

– it is assumed that the pattern of climate influence from both direct temperature effects (primarily at high latitudes) and precipitation changes (at lower latitudes, see Nemani et al. (2003)) will be unchanged through this century

In addition (Enting, 2011) we need to be consistent as to whether any extrapolation is of trends that already included the effects of climate-to-carbon feedbacks. Departures from such extrapolation may simply reflect a failure of these assumptions, rather

than implying the onset of significant non-linearity. Of course, as illustrated in Figure 2, the $1/(1+\beta)$ factor already shows significant non-linearity.

We have not conducted a detailed statistical analysis of our estimates since standard regression formalisms will be inappropriate. Our transformation of the temperature data means that we are fitting time series values that are not independent. The effect of these transformations could be determined but it must be realised that the nature of paleo-climatic reconstructions

means that the temperature data from which we start are not independent. Overall, many aspects of our set of estimates suggest that, in spite of a highly parameterised form of $H(t)$ that reduces our estimation to a single parameter $\gamma'$, the estimation is still poorly conditioned.

We have not attempted to analyse the most striking example of climatic influence on the carbon cycle — the glacial/interglacial changes with $\approx 100$ka periodicity. We have to regard this as beyond the scope of the present paper. Both the consistent peak-

to-peak ranges (in response to complex orbital forcing) and the asymmetry in time, argue for a significant role of non-linear processes such as ecosystem succession and ice-sheet interactions.

Clearly we have to acknowledge that a linear analysis will have a limited domain of applicability even on decadal to century timescales. However, as emphasised by Gloor et al. (2010) an understanding of the linear case is important for providing a reference for identifying non-linear behaviour when it occurs. This paper demonstrates the importance of linear analysis in

clarifying the role of linear estimation.





*Code and data availability.* For archived data for $CO_2$ and the temperature reconstructions see Rayner (2016). This also includes an early version (pre-industrial period only) of the R code used here. Full data sets and code for calculations presented here are available as a zip file in a Figshare project of Ian Enting (Enting, 2019).

**Appendix A: Notation**

The 'hat' notation denotes statistical estimates.

$b_m$   Scaling factor for global temperature signal in region $m$.

$F_{\mathbf{other}}(t)$   Radiative forcing from non-$CO_2$ constituents. (W/m$^2$).

$f_{\mathbf{other}}(p)$   Laplace transform of $F_{\mathrm{other}}(t)$. (W.yr/m$^2$).

$g, g(p)$   Gain in the carbon-climate feedback loop.

$H(t)$   Response function relating warming to $CO_2$ emissions. (PgC/yr$^2$/K)

$H_{\mathbf{ref}}(t)$   Normalised $H$, such that $H(t) = \gamma' H_{\mathrm{ref}}(t)$

$h(p)$   Laplace transform of $H(t)$. (PgC/K/yr).

$k$   Index for functions used by Bauska et al. (2015).

$m$   Index for decomposition of climate-to-carbon influence (see appendix).

$M$   Mass of carbon in atmosphere. (PgC).

$p$   Laplace transform variable (in years$^{-1}$).

$Q(t)$   Departure of atmospheric $CO_2$ from pre-industrial equilibrium. (PgC).

$q(p)$   Laplace transform of $Q(t)$. (PgC yr).

$R(t)$   Response function for $CO_2$.

$r(p)$   Laplace transform of $R(t)$. (yr).

$R_{\mathbf{FB}}(t)$   Response function for $CO_2$ with carbon-climate feedback loop included.

$r_{\mathbf{FB}}(p)$   Laplace transform of $R_{\mathrm{FB}}(t)$. (yr).

$R_{\mathbf{ref}}(t)$   Instance of response used in analysis.

$S(t)$   Anthropogenic $CO_2$ emissions. (PgC/yr).

$s(p)$   Laplace transform of $S(t)$. (PgC).

$T$   Temperature. (K).

$t$   Time. (Years).

$U(t)$   Response function relating radiative forcing to warming. (K/Wm$^{-2}$/yr).

$u(p)$   Laplace transform of $U(t)$. (K/Wm$^{-2}$).

$W(t)$   Warming above notional equilibrium. (K).



$w(p)$ Laplace transform of $W(t)$. (K yr).

$W_m$ Warming above notional equilibrium for region $m$. (K).

$\bar{W}$ Globally coherent temperature change. (K).

$W'_m$ Regional departure from scaled global signal. $W'_m = W_m - b_m \bar{W}$. (K).

$X_k$ Smoothing functions (with $e$-folding time $\lambda_k^{-1}$) (Bauska et al., 2015). (yr$^{-1}$).

$x_k(p)$ Laplace transform of $X_k(t)$.

$\alpha$ Factor describing the linearised forcing dependence of $CO_2$. (W/m$^2$/PgC).

$\beta_{\mathbf{L}}, \beta_{\mathbf{O}}$ Factors defining the partitioning of emissions between atmosphere, biosphere and oceans in the ratio $1 : \beta_{\mathrm{L}} : \beta_{\mathrm{O}}$ (Oeschger
     et al., 1980).

$\beta_k$ Coefficients of temperatures smoothed by $X_k$ (Bauska et al., 2015). (PgC/K).

$\gamma_{\mathbf{L}}, \gamma_{\mathbf{O}}$ Factors specifying the change in terrestrial and oceanic carbon respectively due to change in temperature (Friedlingstein
     et al., 2003). (PgC/K).

$\gamma = \gamma_{\mathrm{L}} + \gamma_{\mathrm{O}}$. (PgC/K).

$\gamma_m$ Factor specifying the change in carbon due to change in temperature in region $m$. (PgC/K).

$\gamma'$ Instantaneous response of $CO_2$ flux to temperature. (PgC/yr/K).

$\psi, \psi(p)$ Airborne fraction of $CO_2$ emissions, as function of timescale, ( i.e. in terms of growth rates (Gloor et al., 2010)).

$\theta_0$ Regression coefficients. Describes offset in pre-industrial $Q(t)$ and $W(t)$.

$\theta_s$ Regression coefficients. Dimensionless.

$\theta_\gamma$ Regression coefficients. $\hat{\theta}_\gamma$ is an estimate of $\gamma'$.

$\lambda_{\mathbf{f}}$ Inverse timescale of forcing. The reciprocal of the forcing timescale $\tau_{\mathrm{f}}$ used by Gloor et al. (2010). (yr$^{-1}$).

$\lambda_k$ Inverse time-scale in $X_k$ (Bauska et al., 2015). (yr$^{-1}$).

$\tau$ Relaxation timescale for temperature-sensitive carbon loss. (yr).

**Appendix B: Disaggregation**

In order to better understand the assumptions underlying empirical estimation of $\gamma$, we need to address the implications of
using a disparate set of regional temperature reconstructions. To this end, we decompose the response to temperature into
regions indexed by $m$. This framework can include land/ocean partitioning as one particular class of disaggregation.

     For any particular timescale, we can write the response as

$$Q = \psi \sum_m \gamma_m W_m$$

If all the regions experience a common temperature perturbation then the $CO_2$ response is characterised by $\gamma = \sum_m \gamma_m$.





More generally, in the spirit of the pattern scaling approximation (Mitchell, 2003) we consider a partitioning

$$W_m = b_m \bar{W} + W'_m$$

where a coherent global temperature signal $\bar{W}$ is manifested in region $m$ with a scale factor $b_m$, and $W'_m$ is the regional departure from $b_m \bar{W}$. The $b_m$ define a pattern of relative responses to a global influence. The problem that confronts us in estimating $\gamma$ is that for different influences such as greenhouse gases, solar variations or large scale volcanic activity the patterns or 'fingerprints' of climate change will differ. In addition, different aspects of climate change will be relevant for climate-to-carbon influences in different locations. Nemani et al. (2003) have presented a partitioning of how plant growth is constrained by temperature vs. water vs. radiation, with temperature being the primary constraint mainly at high northern latitudes.

Special cases of the estimation are:

**global signal** This corresponds to the case where all the $W'_m$ are small. If all the $b_m \approx 1$ (or the $b_m$ take other known values), then the sum $\sum_m b_m \gamma_m$ can be estimated using **any one** of the regional temperature series. However, the individual $\gamma_m$ will not be distinguishable.

**disparate regional signals** In this case the $W'_m$ are sufficiently large and sufficiently distinct to enable the various $\gamma_m$ to be estimated individually, with sufficient data.

In practice we can expect many estimation studies to lie somewhere between these cases. Thus the multivariate fit by Bauska et al. (2015) found a degree of multi-collinearity in the temperature reconstructions, but could estimate some statistically significant regional responses for some regions.

As a Laplace transform, the disaggregation of the climate-to-carbon influence can be written:

$$q(p) - r(p)s(p) = \psi(p) \sum_m \gamma_m(p) \, w_m(p)$$
$$= \psi(p) \sum_m \gamma_m(p) \left[ b_m \bar{w}_m(p) + w'_m(p) \right] \tag{B1}$$

These considerations have an important bearing on our estimation. The paleo-temperature reconstructions are based on small samples from the target regions. Denoting the sampling region by $m'$, it will be seen that $\gamma$ estimates based on temperatures from that region will correspond to

$$\hat{\gamma} = \sum_m \gamma_m(p) \, b_m \Big/ b_{m'} \tag{B2}$$

As well as being applicable to distinguishing between land and ocean exchanges, this general formalism can apply to other disaggregation of processes, e.g. the influence of both temperature and precipitation.

*Author contributions.* Both authors worked on the development of these ideas over many years, and both contributed to the development of the R code. The first author is responsible for the Laplace transform section and for writing the first draft of the manuscript.





*Competing interests.* None

*Acknowledgements.* The first author wishes to acknowledge valuable discussions (in some cases going back over decades) with David Etheridge, Cathy Trudinger, Peter Rayner, Roger Bodman, Mauro Rubino and Xuanze Zhang. The observational temperature record is from the Climate Research Unit, University of East Anglia. The second author gratefully acknowledges support from the Australian Research Council under the Future Fellowship program project number FT130100972).





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





## Figures and tables

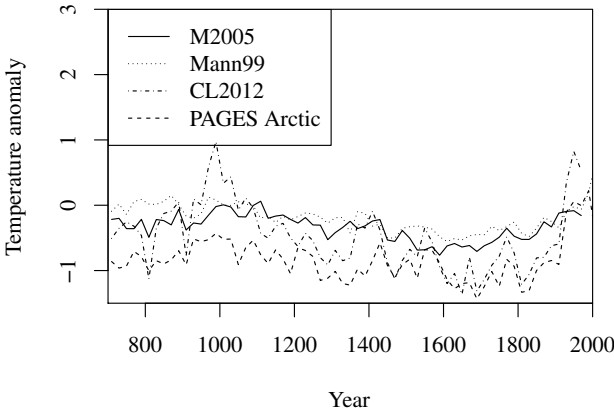

**Figure 1.** Temperature anomalies from various reconstructions from 700–2000, plotted as 20 year running means.

**Table 1.** Estimates, $\hat{\theta}_\gamma$, from pre-industrial fits, extending Rubino et al. (2016). Using $R_{\text{init}}$. The bracketed estimates are potentially influenced by end effects.

| $t_1$ | $t_2$ | Mann | C&L | Arctic | Moberg |
|---|---|---|---|---|---|
| 500 | 1750 | — | (0.216) | (0.482) | (0.605) |
| 1000 | 1750 | (1.105) | 0.290 | 0.637 | 0.705 |
| 1300 | 1750 | 1.856 | 0.659 | 0.625 | 0.986 |
| 1500 | 1800 | 2.191 | 0.806 | 0.883 | 0.992 |

**Table 2.** Estimates $\hat{\theta}_\gamma$, (upper row) and $\hat{\theta}_s$ (lower row) from fits that include the industrial period. Using $R_{\text{init}}$.

| $t_1$ | $t_2$ | Mann | C&L | Arctic | Moberg |
|---|---|---|---|---|---|
| 1000 | 1978 | 1.115 | 0.073 | 0.200 | 1.389 |
|  |  | 0.918 | 0.982 | 0.949 | 0.863 |
| 1500 | 1978 | 2.251 | 0.043 | 0.171 | 2.365 |
|  |  | 0.814 | 0.997 | 0.958 | 0.729 |
| 1800 | 1978 | 1.812 | 0.410 | 0.818 | 2.909 |
|  |  | 0.803 | 0.785 | 0.705 | 0.680 |





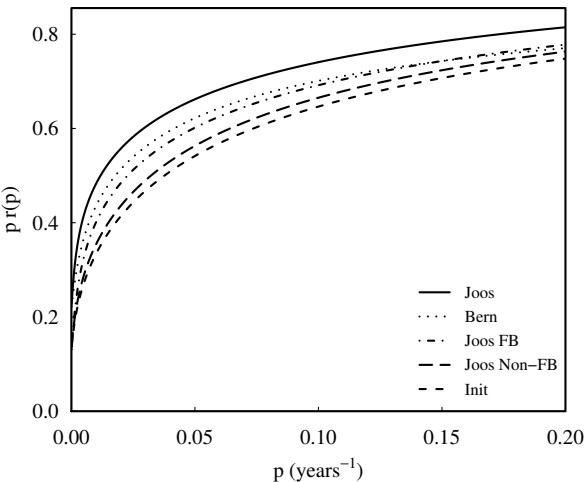

**Figure 2.** The functions $pr(p)$ for various estimates of response functions. We use a number of response functions: $R_{\mathrm{init}}$ from Enting et al. (1994); $R_{\mathrm{Bern}}$ model (as used for GWP definition in IPCC AR4 report (p213)); $R_{\mathrm{Joos}}$ Table 5 from Joos et al. (2013); $R_{\mathrm{FB}}$ from Joos et al. (2013) (table S2 of supplementary information); $R_{\mathrm{no\text{-}FB}}$ from Joos et al. (2013) (table S2 of supplementary information).

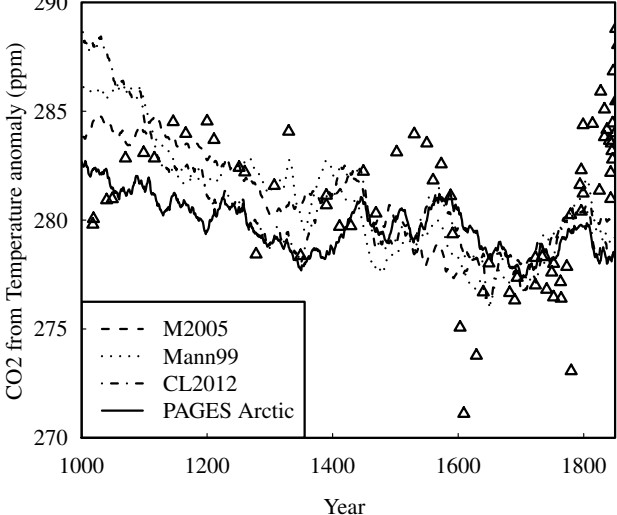

**Figure 3.** Fit to 1300–1750, showing extent to which early $CO_2$ increases could be attributed to temperature changes. See table 1.

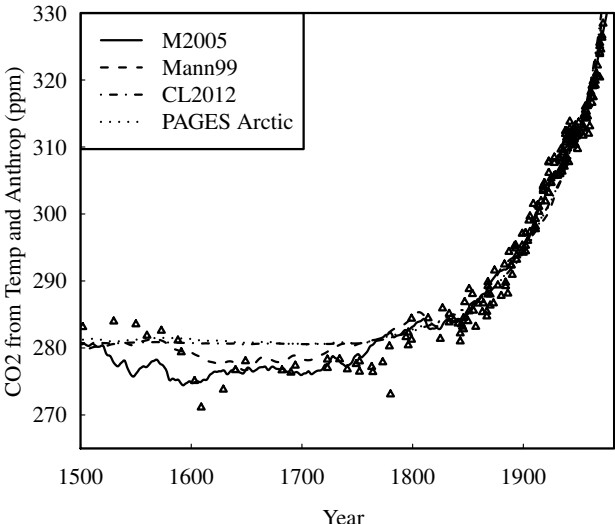

**Figure 4.** Fit to 1500–1978, See table 2.

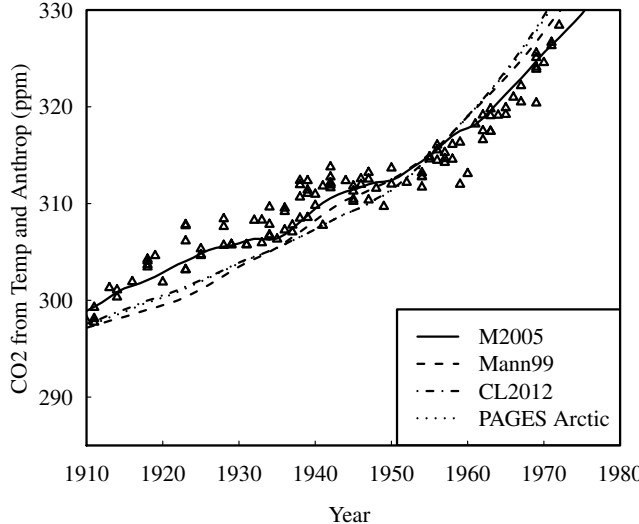

**Figure 5.** Detail for 20th century of Fit to 1500–1978 shown in Figure 4 (Estimates from table 2). The Moberg fit reproduces plateaus in the ice-core record that have hitherto been hard to explain.





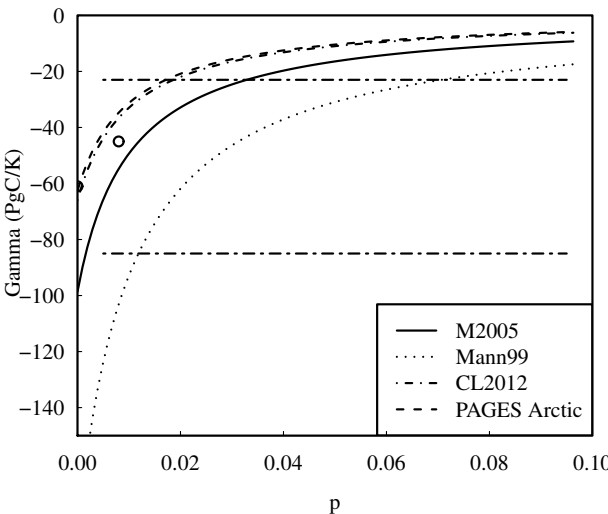

**Figure 6.** Some of our estimates of $\gamma(p) = \hat{\theta}_\gamma/(p + 1/\tau)$ related to other studies. The line type indicates the data sets from which these are derived. All estimates use $\tau = 100$ years but do not include a $(1 - g(p))$ correction. The circle shows our re-interpretation of the $\beta_k$ from Bauska et al. (2015) $\gamma(p) = \beta_k x_k(p)$. The horizontal lines show the two $\gamma$ estimates from Frank et al. (2010).

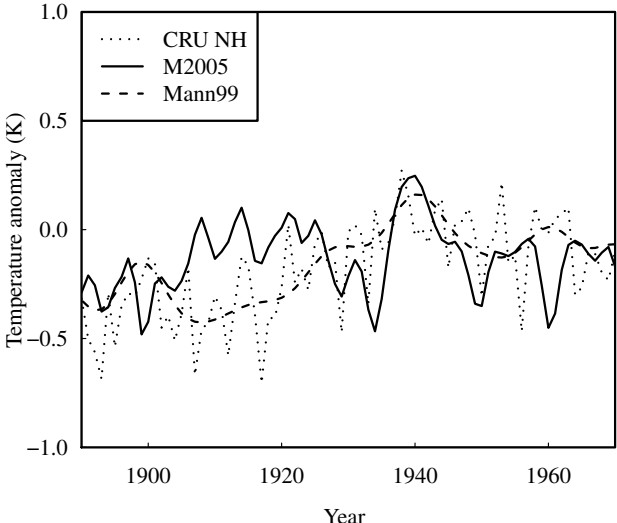

**Figure 7.** Temperature anomalies for 1890–1970 from observational data (CRU NH), and Moberg and Mann reconstructions.