# Peer review of "Estimates of climatic influence on the carbon cycle"

_Earth System Dynamics, 2019_

## Referee Comment (RC1) · Anonymous Referee #1 · 5 Oct 2019

Review of "Estimates of climatic influence on the carbon cycle" by Enting and Clisby

This is a short review due to me not being able to follow the details of the mathematical derivation of the analysis. I follow the general direction, but not the detail of the transforms involved. The equations underpin the whole analysis so this does represent a limit to my ability to perform a thorough review. I will leave it to the editor whether this represents a failing of the reviewer or the paper! I feel that there is clearly a place for detailed mathematical analysis – this is a bedrock of science and we should not be forced to dumb-down important work. But at the same time, some journals are written for a more general science audience who would want to follow the work – ESD is aimed at cross-disciplinary and not all readers have such a strong maths background. Maybe this paper fits better in more specialised journals? Again – a matter for author

and editor to discuss

Overall I support the main message of the paper that climate-carbon feedbacks are not independent of timescales. I think this is really important to stress and to understand. We have known this for some while (e.g. Gregory et al 2009), but additional evidence and discussion is useful.

However, when trying to follow the derivation I got stuck at equation 4 which appears to me to include both anthropogenic emissions and "emissions due to warming". I do not understand this choice. I would have expected to just see "S" here as the anthropogenic emission. Or otherwise to include _both_ the warming and CO2 feedback terms. But to include one of them – the warming term – and not the other appears odd.

Other aspects I would commend:

- the use of multiple datasets of the warming in the paleo record is really nice to see. We know such reconstructions have uncertainties, so to explore implications of this is very good.

- Extension from these periods into the historical is a nice novel analysis

- The conclusion that gamma is not constant for different periods is important – a follow-on conclusion therefore is that one cannot simply translate a measured/inferred gamma from past climates onto future. i.e. the analysis of Frank et al who compared the paleo gamma to model-simulated 21st century is not a valid comparison – rather we need to make sure the models try to simulate the same period as the observations to enable a one-to-one comparison.
* * *

---

## Referee Comment (RC2) · Anonymous Referee #2 · 7 Oct 2019

This paper addresses the important topic of using CO2 and temperature data of the past few thousand years to reconstruct carbon climate feedbacks, and adds to previous work by adopting a more complete formal framework and looking at the ability of same parameter sets to fit different time periods. The most important new messages involve the finding that the climatic controls in CO2 during the Little Ice Age may have been different than more recent periods, and that an understanding of carbon-climate feedbacks may need to consider spatial complexity in the temperature anomalies. The paper builds on the deep insights of the authors and their understanding of nuances of the science. The main issue with this paper is that is such a touch read. I expect as written it will be accessible to only a very limited audience. If the paper is to have significant immediate or lasting value, the presentation needs substantial improvements.

[Figure]

In the detailed comments below, I provide a list of issues that I confronted on reading the paper. While the paper needs line-by-line improvements of this sort, there are also larger issues.

In particular, the derivations using the Laplace transforms seem of secondary importance but are placed front and central, which will discourage many readers. Other than showing a formal equivalent of Eq (1) in frequency space which is not actually used, it seems the main point of the Laplace framing is to flesh out a rigorous distinction between the carbon response function R with and without feedbacks, and to show how the two forms are related. At least, that's about all I got out of it. I thus urge a simplification along these lines: Move much of Section 2 off the main outline, e.g. into an Appendix. Instead, start Section (2) right off with Eq 10, which can be understood without reference to any Laplace transform framing. Also write a new equation that is equivalent to Eq. (10), but replaces carbon pulse response R(t) with R_FB(t). These two equations then cement the idea that there are two slightly different frameworks, depending on whether R contains the climate feedback or not. Here it would be good to point out how the frameworks converge for the preindustrial case. From there, I suggest jumping to the parameterizations used for H(t), also expanding these equations in the time domain, and inserting in to Eq. (10). If I understand correctly, these few equations are sufficient to carry out the main fitting calculations and most of the discussions. If not, I still suggest starting from this basis. With this streamlined outline, the Laplace framing is mainly needed to justify the equivalency of the two versions of Eq. (10) and to show how R_FB is formally related to R(t). These topics would fit fine in an Appendix.

Other general points: If I understand the method correctly, three parameters are fit using CO2 and temperature data: theta_0, theta_s, and theta_gamma in Eq. (17). Another parameter, tau, is not fit, but rather given a seemingly arbitrary value of 100 years. Why this choice? I sense that more discussion is needed on the sensitivity to this choice and why 100 years is justified.

The conclusions need to be rewritten in such a way that the reader isn't forced to remember all the symbol definitions.

Line-by-line comments:

Line 6. Needs rewording. The language is ambiguous as to whether there are discrepancies between different types of temperature records, or whether the carbon feedback parameters optimized on "paleo" data don't fit the early 20th century $CO_2$ record. Also, paleo records are not usually considered relevant for reconstruction of 20th century temperatures. So, on the face of it, this sentence merely states that the paleo temperature reconstructions (e.g. from some earlier period) don't fit the temperature reconstructions of the 20th century, which is akin to saying that the political history of the 19th century was different than that of the 20th century.

8. Would be clearer if this particular reconstruction was named here.

9. Needs rewording. See later comment on line 37.

21: Although Oeschger at al formulate a model to calculate the partitioning of carbon into different reservoirs, I'm quite sure they do not define factors that are direct analogues of the beta factors used here. The context for this citation therefore appears incorrect.

37: This sentence, which is also echoed in the abstract, is hard to follow. The Moberg reconstruction did not use $CO_2$ as an input, but the wording suggests it did. I think the point is that the temperature reconstructions by Moberg do not reflect uniform weighting of land temperature. By chance, it seems that their average is weighted towards regions with larger carbon responses, yielding a larger (or a more appropriate?) $CO_2$ response than would be obtained for an area-weighted hemispheric average.

Eq. (2). The quantity $Q(t)$ (or $q(p)$) needs defining. Alpha*Q is defined to be radiative forcing, but Q itself is not defined here, leaving it to the readers imagination.

Eq. (4). Some additional explanation is needed on how this equation can be applied

separately to both preindustrial and modern situations. Also, for a preindustrial appli-
cation Q(t) is the not the excess above a pre-industrial level, but rather a perturbation
in the preindustrial level itself, right? The text leading into Eq. (4) is thus problem-
atic, i.e. Q(t) needs to be explicitly defined as the perturbation relative to a reference
(i.e. constant) preindustrial level. This should be done on first usage (see above). A
similar clear definition is needed for W(t). (Comment might be partly superseded by
reorganization proposed above).

59: Confusing. If p r(p) is equal to the airborne fraction and psi(p) is also equal to the
airborne fraction, then why not write an equation that sets them equal to each other?
Also, are we talking about the cumulative airborne fraction or the airborne fraction
based on yearly emissions?

64: Hard to follow. I sense it would help to show an additional equation of the sort q(p)
= . . . between (6) and (9) to clarify how g(p) inserts into the math.

73: Partition into what?

90: Meaning of "ambiguity" unclear. I think there is no conceptual ambiguity here.
Rather, the key issue is distinguishing the quantities from past records of variability.

Eq(11) and Eqs (12). Does this formulation have a simple basis in box modelling?
Would be good to clarify. I notice that this formulation fixes the ratio of the responses at
the high and low frequency limits. Why is this reasonable? Reading ahead, I see that
Eq. (15) somewhat answered these questions. This suggests flipping the order and
putting Eq. (15) ahead of (11) and (12) to help justify the approach. Also would be good
to clarify whether H(t) has units of flux or units of amount, i.e. does a step change in
temperature immediately release a finite pulse or does it cause a step change in flux?
(This comment might be superseded if text is reorganized as proposed above).

Eq. (16). This would fit better in the next section, see below.

Eq (17). Hard to follow. What is mean by "normalization" here? What do Rref and

Href represent? Why is the "ref" subscript needed at all? The need for theta_0 is explained. But the need for theta_s and theta_gamma is not. What happened to gamma_prime? What parameters are adjusted in the fitting process? What are the data inputs? What is being minimized? If correct, it would help clarify to explicitly write: R(t) = theta_s*Rref(t), H(t) = thetma_gamma*Href(t), where Rref is . . . etc.

124: There is no hat symbol appearing in Eq. (17). The hat symbol should be defined only when first used.

Eq. (18). Why the jump back to a Laplace transform formulation? Is the regression actually done in p space? If not, why not stick to the time domain? It's not obvious how Eq. 18 is derived. Which previous equations are combined used under what mathematical assumptions? In this context it would be good to clarify why a "preindustrial" situation allows a term to be eliminated. This later point would be covered by addressing the earlier query about Eq. (4).

139. Hard to follow because the relationship between gamma' and theta_gamma has not yet been clarified. If they are equal to each other, why the need for two symbols? Eq. (19), Similar to Eq. (18), an explanation is needed for how to derive Eq (19). What Equations are combined?

208. This text could also use some expansion, similar to the related point in the Introduction. The relevant regions are those in which carbon fluxes are particularly sensitive to temperature.

209. Actually, Rafelski's model does partly account for the plateau.

238 The meaning of "correspond" is unclear.

240 Better to put the xk(p) above when introducing Xk(t).

255 having

261. The dash is confusing. Replace with "and"?

262. "notably" is dangling.

282-287. I am not overly concerned about the lack of error analysis, but the difficulty arising from non-independence could be overcome with an appropriate error model that accounted for the dependence.

287-291. Could be cut as out of scope.

300: I see no "hats" on the list that follows. This comment belongs wherever the hat is first actually used.

Table 1 and Table 2. Meaning of t1 and t2 not obvious at first glance. I suggest changing the t1 and t2 headers to "time frame", and collapse t1 and t2 columns to one column, e.g. 500-1750. Also, don't the numbers depend on the choice if tau? The table should clarify this.

Figure 3. Ice-core CO2 data source is not specified.

---

## Author Comment (AC1) · 7 Oct 2019

Ian Enting and Nathan Clisby

ian.g.enting@gmail.com

On the basis of presentations and discussions at the AGU Chapman Conference on Understanding Carbon Climate Feedbacks, we propose the following changes to notation to make our formalism more directly comparable to earlier work:

(1) $\alpha \to \alpha'$, so that the radiative forcing from a $CO_2$ perturbation, $q(p)$, becomes $\alpha' q(p)$

(2) use $\alpha(p)$ to denote $\alpha' u(p)$ so that $\alpha(p)$ is the time-scale dependent generalisation of $\alpha$ as used by Friedlingstein et al., i.e. the temperature response to a $CO_2$ perturbation, $q(p)$, is $\alpha(p) q(p)$.

---

## Short Comment (SC1) · 21 Oct 2019

In our paper, equation (4) follows directly from equation (3) by using the relation $p\,r(p) = 1/(1 + \beta(p))$ since these are both expressions for the airborne fraction (for growth with timescale $p$ ).

Equation (4) is also a direct generalisation of the relation which was defined by Friedlingstein et al. (2004) and is used extensively in interpretation of modelling calculations.

Apart from its relation to a widely-used formalism, in our view it is a reasonable process-oriented approach for describing the system. However, the formalism is less directly related to what can be observed. The relation between the process description and

observations is the reason for some of the mathematical complexity.

Referee 2 is entirely correct in noting that we work from equation (3) and make no use of equation (4). However, the vast majority of studies of climate-to-carbon feedback are framed in terms of a restricted form of equation (4) and so we think it is important to note the connection.

One can factor in the warming from industrial $CO_2$ emissions and reduce equation (4) to the special case

$$q(p) = \frac{s(p)/p}{[1 + \beta(p)][1 - g(p)]}$$

However, this only applies if ALL the temperature variation is due to anthropogenic $CO_2$ emissions. We think that this assumption is unjustified. Our analysis of the pre-industrial period assumes that NONE of the pre-industrial temperature change and $CO_2$ variation was due to anthropogenic emissions. (Other studies have proposed a pre-industrial anthropogenic influence. The formalism of equation (4) could be used to explore such proposals.)

For the early 20th century, we want to keep open the possibility that some of the warming may not be due to $CO_2$ and so keep the terms in equation (4) separate. (There is, of course a contrarian view that almost none of the 20th century warming is due to anthropogenic $CO_2$ and merely represents a recovery from the Little Ice Age (LIA) and even that the $CO_2$ increase is due to the warming and not due to anthropogenic emissions. In terms of our formalism, this would require very small $r(p)$ and very large $\gamma$.)

Combining the direct and indirect consequences of $CO_2$ emissions as is done above, has the effect of replacing the $CO_2$ response $r(p)$ by a rescaled response $r_{FB}(p) = r(p)/(1 - g(p))$. An old analysis by Enting (1992) showed that there is no possible response function that could explain $CO_2$ growth over the industrial period as a linear response to anthropogenic emissions, ruling out such a response $r_{FB}$ as the SOLE

contribution to $CO_2$ changes. (Possible was defined by the response, in the time domain, being always positive, the derivative being always negative and the second derivative being positive.) The conclusion was that some of the $CO_2$ increase was the effect of warming during recovery from the LIA.

I.G. Enting. (1992). The incompatibility of ice-core $CO_2$ data with reconstructions of biotic $CO_2$ sources. II. The influence of $CO_2$-fertilised growth. *Tellus* **44B**, 23–32.

---

## Author Comment (AC3) · 13 Nov 2019

**Note** This author comment on Laplace transform is posted to provide context for detailed reply to review comments.

We have found the Laplace transform has been a useful way of exploring relations between components of the carbon cycle. However, our main reason for introducing the Laplace transform in our paper is as a way of emphasising the relation

$$p\,r(p) = \frac{1}{1 + \beta_O(p) + \beta_L(p)} \qquad (AC3.1)$$

which links our approach in terms of response functions to the description in terms of $\beta$ and $\gamma$ which has been used in the majority of studies of climate to carbon influence:

$$q(p) = \frac{s(p)/p}{1 + \beta_O(p) + \beta_L(p)} + \frac{\gamma_O(p) + \gamma_L(p)}{1 + \beta_O(p) + \beta_L(p)} \qquad \text{(Enting and Clisby (5))}$$

This maps one-to-one as a generalisation of the relation given by Friedlingstein et al (2003) (see equation (1) of Enting and Clisby) which in turn has defined the concepts and notation used in the majority of studies of climate-to-carbon feedback.

As summarised below, the Laplace transform captures both the asymptotic form of the airborne fraction and also the relations between the $\beta$ factors and reservoir-specific response functions used in much earlier work, in particular the characterisation of ocean carbon models. Similarly, the Laplace transform provides a compact way of showing how our estimates relate to the weighted averaging used by Bauska et al. (2015). Our final response to review comments will indicate how we propose to revise our paper in order to emphasise these issues.

It is well known that in a linear system, subject to exponential forcing, all the system components will respond with the same exponential growth rate. The results from Oeschger et al (1980) cited in AC2 are a special case of this. An exponential response at time $t$ requires exponential forcing over all times $t' \in (-\infty, t]$.

If the system behaviour is described using a response function $R(t)$, forcing $S(t) = A \exp(\lambda t)$ leads to response

$$Q(t) = A \int_{-\infty}^{t} \exp(\lambda t') R(t - t') dt' \qquad (AC3.2)$$

or putting $t'' = t - t'$

$$Q(t) = A \exp(\lambda t) \int_{0}^{\infty} \exp(-\lambda t'') R(t'') dt'' = A \exp(\lambda t) r(\lambda) \qquad (AC3.3)$$

Thus the instantaneous airborne fraction is given by

$$\dot{Q}(t)/S(t) = \lambda\,r(\lambda)$$

and similarly the cumulative airborne fraction is

$$Q(t) \left/ \int_{-\infty}^{t} S(t')\,dt' \right. = \lambda\,r(\lambda)$$

This is the asymptotic limit of Laplace transform relations which apply for functions defined on $[0, \infty)$, where for an exponential response function $\exp(-\alpha t)$, the Laplace transform gives

$$q(p) = \frac{A}{p - \lambda} \times \frac{1}{p + \alpha} = \frac{A}{\alpha - \lambda} \left[ \frac{1}{p - \lambda} + \frac{1}{p + \alpha} \right] \qquad (AC3.4)$$

which has limit as per eqn (AC3.3) plus a transient term decaying as $\exp(-\alpha t)$.

This quantifies role of transients, as discussed by Gloor et al. (2010). The generalisation to when $R(t)$ is expressed as a sum of exponentials is obvious. (The special case of term with $\alpha = 0$ redefines the origin of $Q$).

As noted above (and our comment SC1) relation (AC3.1) above is not widely recognised. Thus reviewer 2 did not recognise the relation between the $\beta$ factors and the relation given by Oeschger et al (1980). Similarly in the comprehensive review of feedbacks by Gregory (2009), the abstract states that "The concentration to carbon feedback is negative, it has generally received less attention in the literature ..." [i.e. less compared to the climate-carbon feedback].

In reality, as $R(t)$, the concentration-to-carbon feedback is widely discussed as representing carbon cycle response in carbon cycle studies. $R(t)$ is also very widely discussed because it defines the reference level for GWP. (Note that more recent usage has $R_{FB}(t)$ rather than $R(t)$ in the definition of GWP).

[Figure]

Of course, all the relations that we express as Laplace transforms can be expressed using integro-differential equations in the time domain. We would argue that actually doing so would deter readers more than the use of Laplace transforms.

Assuming that one wants to avoid the use of infinite sums of successively higher order integrals (representing binomial expansions of denominators of Laplace transform expressions), the fractions have to be multiplied out and equation (4) of our paper becomes:

$$Q(t) + \int_0^t Q(t')[B_O(t-t') + B_L(t-t')]\,dt' = \int_0^t S(t')\,dt' - \int_0^t W(t')[\Gamma_O(t-t') + \Gamma_L(t-t')]\,dt'$$
$$(AC3.5)$$

where $B_O(t)$, $B_L(t)$, $\Gamma_O(t)$, $\Gamma_L(t)$ are response functions, with Laplace transforms $\beta_O(p), \beta_L(p), \gamma_O(p), \gamma_L(p)$ that characterise the feedbacks from concentration and temperature for the ocean and land pairs.

The $B_O(t)$, $B_L(t)$ that describe responses to concentration, can be related to responses $R_O(t)$, $R_L(t)$ that describe responses to fluxes. Here $R_O(t)$ is an ocean-only response (e.g. as used by Oeschger and Heimann (1983) and many subsequent studies) and $R_L(t)$ is a biota-only response (e.g. as calculated by Friedlingstein and reported in Enting et al (1994)).

In terms of Laplace transforms, the connection is (see eqn (13b) of Enting (2007)):

$$\beta_O = \frac{1}{pr_O(p)} - 1$$

and

$$\beta_L = \frac{1}{pr_L(p)} - 1$$

Thus in the time domain

$$\int_0^t R_L(t-t')\Phi(t')dt' + \int_0^t R_L(t-t') \int_0^{t'} B_L(t-t'')\Phi(t'')dt'' = \int_0^t \Phi(t')dt' \quad (AC3.6)$$

serves to connect the response calculated by Friedlingstein in 1994 to the $\beta$ factor introduced by Friedlingstein et al in 2003. (An obvious correspondence relates the $B_O(t)$ (and thus its Laplace transform $\beta_O$) to the ocean-only response $R_O(t)$ used by Oeschger and Heimann (1983) and in many later studies.)

We would argue that relations such as (AC3.5) and (AC3.6) are much more comprehensible as Laplace transforms.

**Additional reference (for this comment, not proposed for paper)**

Oeschger, H. and Heimann, M. (1983). Uncertainties of predictions of future atmospheric $CO_2$ concentrations. *J. Geophys Res.*, **883**, 1258–1262.

---

## Author Comment (AC4) · 15 Nov 2019

Ian Enting and Nathan Clisby

ian.g.enting@gmail.com

**Line-by-line comments**.
This includes responses to all line-by-line comments from reviewer 2, and some additional changes to address issues raised in the review.

**Additional change: Line 3** We feel that inserting a summary of the technique at this point in the abstract can clarify issues noted in the next few review comments, and also insert the sentence into the summary of the paper in section 1. However we suggest that this sentence be part of a largely restructured abstract as proposed below.
*Proposed change. Insert:* We propose a parameterised relation between temperature and $CO_2$ changes and use regression analysis to estimate the strength of the climateto-carbon influence by matching calculated $CO_2$ levels to measured concentrations obtained from ice core data.

**Line 6** *Proposed change* See proposed revision of whole abstract.

**Line 8** Constraint is not being able to have citations in abstract. *Proposed change* None

**Line 9** See our comments for line 37. We think the issue is addressed by our proposed insertion at line 3. *Proposed change* (see above).

**Line 21** See author comment number 2. *Proposed change* Move citation of Oeschger et al. to after equation (5).

**Additional change: After line 26** To emphasise the points that we make in author comment 3. *Proposed change. Insert:* Following Rubino et al. (2016) we note that equation (1) generalises to a timescale-dependent form and that, as shown by equation (5), the relation is particularly clear when using Laplace transforms. Since we are analysing pre-industrial variation, we cannot talk about unique pre-industrial values of either $CO_2$ or temperature. Rather, we have to consider $Q(t)$ and $W(t)$ as perturbations about arbitrary reference levels with a requirement for consistency at a notional equilibrium state that may never have actually occurred.

**Additional change: Line 30** *Proposed change, insert sentence* We use a parameterised relation between temperature and $CO_2$ changes and apply regression analysis to estimate the strength of the climate-to-carbon influence by matching calculated $CO_2$ levels to measured concentrations obtained from ice core data.

[Figure]

**Line 37** We think that the proposed change at line 30 should dispel any implication that the paleo-temperature reconstructions are determined by $CO_2$ *Proposed change* as above

**Eqn 2** $Q(t)$ has already been defined in connection with equation (1). However, to deal with the broader issue (which we had deferred until later): *Proposed change* Enting (2010) analysed the carbon-climate feedback loop in terms of $CO_2$ variations, $Q(t)$, and temperature variations, $W(t)$, (with Laplace transforms $q(p)$ and $w(p)$ respectively) as

$$w(p) = u(p)[\alpha' \, q(p) + f_{\text{other}}(p)] \qquad \text{(Enting and Clisby(2))}$$

where $U(t)$ (with transform $u(p)$) gives the temperature response to radiative forcing. Given the variations, we cannot specify unique pre-industrial values of either $CO_2$ or temperature. Rather they are defined relative to reference levels, subject to a consistency constraint that depends on the strength of the climate-to-carbon coupling and which becomes part of our estimation.

**Additional change: Eqn (3) and following** As foreshadowed in author comment 1. Introduce $\alpha' u(p) = \alpha(p)$ for greater consistency with other studies. *Proposed change*

$$w(p) = \alpha(p)q(p) + w_{\text{other}} \qquad \text{(Enting and Clisby(3))}$$

where we define $\alpha(p) = \alpha' \, u(p)$ so that $\alpha(p)$ becomes a timescale-dependent generalisation of the term $\alpha$ used in other studies such as Friedlingstein et al. (2006: eqn 6) and subsequent analysis.

**Eqn 4** *Proposed change* As indicated in connection with eqn (2).

**Line 59** Since the Laplace transform expression $p\,r(p)$ is giving the airborne fraction for emissions that grow as $\exp(pt)$, cumulative and current airborne fractions are equal. *Proposed change* As noted by Oeschger et al. (1980) the partition factors depend on the timescale of growth and need to be written as $\beta_O(p)$ and $\beta_L(p)$. This can be related to response functions (Enting, 1990) so that for exponential growth at rate $p$ (when cumulative and instantaneous airborne fractions are equal), the airborne fraction is $\psi(p) = 1/(1 + \beta_l(p) + \beta_O(p)) = p\,r(p)$. Defining $\gamma(p) = -h(p)/p$ and noting that $s(p)/p$ is the transform of integrated emissions, relation (4) becomes

**Line 64** Of course all the connection is an example of normal feedback analysis and the specific case is given by Enting (2010) and Gregory et al (2009). But since Gregory doesn't do the timescale dependence and Enting (2010) may be difficult to access, we propose. (Note that, as foreshadowed in author comment 1, we have replaced $\alpha$ by $\alpha'$ and defined $\alpha(p) = \alpha'\,u(p)$). *Proposed change: replace line 63 with* Substituting (3) into (4) gives

$$q(p) = r(p)[s(p) + \alpha(p)h(p)q(p) + h(p)w_{other}(p)] \qquad (new)$$

whence

$$q(p)[1 - \alpha(p)r(p)h(p)] = r(p)[s(p) + h(p)w_{other}(p)] \qquad (new)$$

so that the feedback loop has a gain

**Additional change: Equations 6, 7, 8** *Proposed change* Renumber these (and all subsequent equations) and replace $\alpha\,u(p)$ by $\alpha(p)$.

**Line 73** *Proposed change* In the absence of any ability to partition the temperature changes represented by $w(p)$ into changes caused by $CO_2$ and changes from other causes, relation (9) . . .

**Line 93 (reviewer has this as line 90)** Ambiguity referred to past studies where the (now unambiguous) distinction had not been made. *Proposed change* A new part of

our analysis is recognising the uncertainties in using responses that may explicitly represent $R_{\text{FB}}$ explicitly vs responses presented as $R(t)$ but which may implicitly include climate-carbon feedback effects and estimating a scale factor to account for these uncertainties.

**Eqn 11,12** We are not using the limits to fix the behaviour. We are merely saying what the limits are, in order to facilitate comparisons with other studies. The word 'interpolates' is a description of what the function does, not how we constructed it. Our re-wording also addresses issues relevant to the choice of $\tau$. *Proposed change* Parameterisation (11) has simple limits of constant $h$ and constant $\gamma$ at large and small values of $p$ respectively. However neither of these limits will be of great importance for our analysis, given the ice core data span less than 2000 years, and the nature of the bubble trapping smooths out rapid variations in concentration.

**Additional change: After Line 114** Reviewer 2 notes the need to discuss the choice of $\tau$, without specifying a particular place to do so. *Proposed change* As in the studies by Rubino et al. (2016) we have used $\tau = 100$ years, guided by the behaviour of various carbon cycle models, and by the recognition that, as noted above, the behaviour of $h(p)$ on timescales that are much longer or much shorter will not greatly influence our analysis.

**Eqn 16** *Proposed change* As suggested, move equation to next section, and modify lines 119–120 as described in next point.

**Additional change: Replacing Lines 119–120**
*Proposed change* Our analysis uses regression based on equation (10), generalising the work of Rubino et al. (2016) who only used the second term. We use the generic

notation $R_{\text{ref}}$ to cover various estimates and estimate a scale factor $\theta_s$. Similarly, we define a reference function $H_{\text{ref}}$ by

$$H(t) = \gamma' H_{\text{ref}}(t) \tag{16}$$

leaving another scale factor, $\gamma'$, to be estimated.

**Eqn 17** We think the issues are addressed by the change above. *Proposed change* As above.

**Line 124** *Proposed change* As suggested, move sentence about hat notation to immediately after equation (18).

**Eqn 18** The discussion continues in terms of Laplace transforms, firstly for compactness and readability (as we note in lines 134), but mainly as a path to discussion $\theta_s$. Laplace transform relations are multiplicative and so we can factor out the terms $r_{\text{FB}}(p)$ and $r(p)$ to suggest $1/\theta_s$ is an estimate of $r_{\text{FB}}(p)/r(p)$ (while also making it explicit that a single number is serving as a proxy for a relation that depends on timescale). Without the ability to factor out terms, working in the time domain would mean that the relations would have to be expressed as integrals involving $W(t)$ and $H(t)$. (see author comment 3). *Proposed change* None/

**Additional change after eqn 18**
*Proposed change insert after eqn 18* as used by Rubino et al. (2016).

**Line 139** If we had a reliable estimate of $R(t)$, then we could dispense with $\theta_\gamma$ and just use $\gamma'$. This assumption was made in the analysis by Rubino et al. (2016). Similarly we could make $\theta_s$ exactly equal to 1 rather than a quantity to be estimated (thus leading

to estimation based on equation (19)). However, as argued (with specific examples) by Enting (2011), empirical estimates of the $CO_2$ response are going to be closer to $R_{FB}$ rather than $R(t)$. This interpretation is confirmed by our finding of estimates of $\theta_s$ that are less than one. Of course, as noted in the previous point, encapsulating the difference between $R(t)$ and $R_{FB}(t)$ through a single scaling factor is only an approximation. *Proposed change* None at this point. We think that the changes made at Line 93 address this issue.

**Line 208** Expanding a little, to be more explicit.
*Proposed change* This suggests that the Moberg data are sampling regions that do not always represent hemispheric averages but which have a larger than average sensitivity of terrestrial carbon to northern hemisphere temperature.

**Line 209** *Proposed change, replace sentence on 208–9 by* This would explain why Bastos et al. (2016) were unable to explain the 1940s plateau and why Rafelski et al. (2009) were unable to explain the combination of the plateau and the anomalously great increase in $CO_2$ over the preceding decades.

**Line 238** *Proposed change* Bauska et al. based their analyses on $CO_2$ fluxes obtained by model-based deconvolutions. Within the linear approximation, deconvolution can be represented as $1/r(p)$ (Enting 2007) and so their analysis corresponds to ...

**Line 240** *Proposed change, move as suggested. So at end of line 236:* The Laplace transform is $x_k(p) = \lambda_k/(p + \lambda_k)$.

**Line 240** Rewording, following on from the change to line 238. *Proposed change* In terms of our parameterisation this corresponds to fitting $\gamma(p)\,w(p) \sim \beta_k\,x_k(p)\,w(p)$

l**Line 255** *Proposed change (as suggested)* having

**Line 261** *Proposed change (remove dash as suggested) but emphasise that failure to include time-scale dependence is a form of over-parameterisation. Change to:* . . . over-parameterisation, especially through the failure . . .

**Line 262** dangling word reflects our incorrect bracketing of reference. *Proposed change* .. notably Bauska et al. (2015).

**Line 282–287** While desirable, an error analysis would require knowing the covariance structure of the paleodata. We see this as beyond the scope of this paper. *Proposed change* None.

**Line 287–291** This could indeed be cut, but we think that our reasons for not including analysis of glacial-interglacial changes are things that should be considered by anyone who extends our analysis to consider pre-Holocene times. *Proposed change:* Defer to editor's view.

**Line 300** The use of the hat as common statistical notation is introduced in line 124, immediately before the section where we start using it. Since it is part of the notation, we think it is worth flagging in the notation section, in the same way as other items in the notation are duplicating definitions that are given in the text. *Proposed change* None.

**Additional change: Line 332** (As foreshadowed in AC1) *Proposed change* Keep this

line as definition of $\alpha'$ and follow with line defining $\alpha(p) = \alpha' \, u(p)$

**Table 1** *Proposed change. Reword first 2 sentences of caption as* Estimates, $\hat{\theta}_\gamma$, from fitting ice core $CO_2$ data over the time frame shown. All cases use response $R_{\text{init}}$ and $\tau = 100$ years

**Table 2** *Proposed change. Reword first 2 sentences of caption as* Estimates, $\hat{\theta}_\gamma$ (upper row), and $\hat{\theta}_s$ (lower row), from fitting ice core $CO_2$ data over the time frame shown. All cases use response $R_{\text{init}}$ and $\tau = 100$ years

**Figure 3** (with same change on figures 4 and 5) *Proposed change, add to caption* Triangles show $CO_2$ concentrations from Law Dome DSS ice core (see Rubino et al. (2016).

**Additional change: Line 384** *Proposed to insert:* The authors gratefully acknowledge the thoughtful and detailed review comments.

**Proposed re-write of abstract**

The influence of climatic change on the carbon cycle is important as part of a $CO_2$-climate feedback loop. Quantitative analysis needs to go beyond characterising the climate-to-$CO_2$ influence by a single number $\gamma$ relating $CO_2$ variation to temperature variation.

Several paleo-temperature reconstructions are analysed assuming that $CO_2$ is influenced by the history of temperature changes, thereby incorporating a time-scale dependence into the characterisation of the climate-to-carbon influence. The analysis is based on a parameterised relation between temperature and $CO_2$ changes and

uses regression analysis to estimate the strength of the climate-to-carbon influence by matching calculated $CO_2$ levels to measured concentrations obtained from ice core data.

Expanding on such previous analyses of the pre-industrial period 1000–1750, shows that even when accounting for time-scale dependence, the coldest part of the Little Ice Age seems to reflect different behaviour to that in earlier or later centuries. Different temperature reconstructions appear to capture distinct aspects of pre-industrial climate fluctuations with varying consequences for implied $CO_2$ changes and differences in when and how closely these match the observed $CO_2$ changes. This disparate behaviour is consistent with recent analyses showing a lack of global coherence in pre-industrial climate variation.

The analysis is extended into the industrial period. both by extrapolating pre-industrial fits and, once taking anthropogenic emissions into account, fits that include the 20th century. Again the results show disparate behaviour from different paleo-data sets. Most paleo-temperature data fail to imply a plateau (or plateaus) in 20th century ice-core $CO_2$. One particular high-resolution reconstruction that does not closely reflect hemispheric temperature changes, does imply $CO_2$ changes that match observed plateaus in concentration. The implication is that the reconstruction possibly sampled a pattern of variation where the terrestrial carbon exchange is anomalously sensitive to regional climate variations. These various results suggest that this type of empirical study may have limited applicability to the 21st century.

---

## Author Comment (AC5) · 29 Nov 2019

**Overall response**

Our response AC3 discusses the importance of the Laplace transform formalism. Apart from being more compact and flexible than the corresponding integro-differential equations in the time domain, we think it is important for 3 reasons:

* Relating our formalism to the framework introduced by Friedlingstein et al (2003) and used in the majority of subsequent studies. (in particular relating the $\beta$ factors to $CO_2$ responses, $R(t)$)

* explaining how our estimates relate to the weighted sums calculated by Bauska et al.

[Figure]

* clarifying the issues of $R(t)$ vs $R_{\mathbf{FB}}$. Reviewer 2 had thought that this is the only use that we had made of Laplace transforms, but arguably it is less important than relating our formalism to previous studies.

Consequently, we feel that it is appropriate to retain the Laplace transform formalism in the body of the paper.

**Reviewer 1.**:
The editor has confirmed that ESD is an appropriate journal for presenting work at this level detail. We are gratified by this, since we feel that in the past some key connections (see comments above, and AC3) have been missed because the relevant mathematics has only been published in more specialised contexts (e.g. Enting et al 1994; Enting 2007, 2010).

**Reviewer 2:**

Our responses to the line-by-line comments from reviewer 2 are given in our comment AC4 which lists our proposed changes. These include additional words to address the issue of the choice of $\tau$.

We thank the reviewer for the care and effort that has gone into the review.

**Additional changes not flagged in AC4**

* We will add a reference to Enting and Mansbridge (1987) when discussing deconvolution in the context of the Bauska analysis. (Rather than referring to Enting (2007), which we had previously flagged we would do in AC4 for line 238.)

* Tables 1 and 2 will be restructured so that the date ranges form a single column, as suggested by reviewer 2.

* Additional wording at beginning of 'Estimation' section in association with following the reviewer's suggestion of moving the equation for $H(t)$.

We are gratified that both reviewers acknowledge the importance of our study.

**Reference (referenced in comments but not in paper)**

I. G. Enting (2007). Laplace transform analysis of the carbon cycle. *Environmental Modelling and Software* **22**:1488–1497.